# The Role of Mitochondria in the Chemoresistance of Pancreatic Cancer Cells

**DOI:** 10.3390/cells10030497

**Published:** 2021-02-25

**Authors:** Yibo Fu, Francesca Ricciardiello, Gang Yang, Jiangdong Qiu, Hua Huang, Jianchun Xiao, Zhe Cao, Fangyu Zhao, Yueze Liu, Wenhao Luo, Guangyu Chen, Lei You, Ferdinando Chiaradonna, Lianfang Zheng, Taiping Zhang

**Affiliations:** 1General Surgery Department, State Key Laboratory of Complex Severe and Rare Diseases, Peking Union Medical College Hospital, Chinese Academy of Medical Sciences and Peking Union Medical College, Beijing 100730, China; fyball@student.pumc.edu.cn (Y.F.); 2010302180268@whu.edu.cn (G.Y.); qjdpumch@163.com (J.Q.); huanghua@pumch.cn (H.H.); xiaojianchun@pumch.cn (J.X.); caozhe@pumch.cn (Z.C.); pumc_fangyuzhao@student.pumc.edu.cn (F.Z.); liuyzpumch@163.com (Y.L.); pumc_luowenhao@student.pumc.edu.cn (W.L.); chengyu0304@hotmail.com (G.C.); florayo@163.com (L.Y.); 2Department of Biotechnology and Bioscience, University of Milano Bicocca, 20126 Milano, Italy; francesca.ricciardiello@unimib.it; 3Department of Nuclear Medicine, Peking Union Medical College Hospital, Chinese Academy of Medical Sciences and Peking Union Medical College, Beijing 100730, China; lianfangzheng@yahoo.com; 4Clinical Immunology Center, Chinese Academy of Medical Sciences and Peking Union Medical College, Beijing 100730, China

**Keywords:** pancreatic cancer, mitochondria, chemoresistance, metabolism, apoptosis

## Abstract

The first-line chemotherapies for patients with unresectable pancreatic cancer (PC) are 5-fluorouracil (5-FU) and gemcitabine therapy. However, due to chemoresistance the prognosis of patients with PC has not been significantly improved. Mitochondria are essential organelles in eukaryotes that evolved from aerobic bacteria. In recent years, many studies have shown that mitochondria play important roles in tumorigenesis and may act as chemotherapeutic targets in PC. In addition, according to recent studies, mitochondria may play important roles in the chemoresistance of PC by affecting apoptosis, metabolism, mtDNA metabolism, and mitochondrial dynamics. Interfering with some of these factors in mitochondria may improve the sensitivity of PC cells to chemotherapeutic agents, such as gemcitabine, making mitochondria promising targets for overcoming chemoresistance in PC.

## 1. Pancreatic Cancer

The greatest contribution of the pancreas is its exocrine function of secreting digestive enzymes into the gut. In contrast, only approximately 1% of pancreatic tissue is involved in endocrine function. The exocrine function of the pancreas is executed by acinar and duct cells that produce digestive enzymes and deliver enzymes into the duodenum [1]. These cells compose the vast majority of pancreatic tissue. From a pathological point of view, acinar and ductal cells are involved in disease processes, and currently acinar and ductal cells are considered the initiating cells for precursor lesions of PC. Importantly, differentiated acinar and ductal cells in the pancreas possess a high degree of cellular plasticity to maintain normal tissue homeostasis and initiate an adequate response to injury under physiological circumstances. Unfortunately, under the permissive influence of oncogenic KRAS and/or other predisposing factors, such as chronic inflammation and epigenetic alterations, this plasticity is also a predisposing factor for malignant transformation in these cells [2,3].

With its poor prognosis, PC has a high mortality rate worldwide, with a 5-year survival rate of only approximately 9% [4,5]. This malignant tumor progresses rapidly, with inconspicuous symptoms during the initial period of development [5]. Although chemotherapy can improve survival rates, surgical resection is the only possible way to cure PC. However, because of the atypical symptoms, rapid progression, and lack of diagnostic markers, most patients’ cancer is unresectable at diagnosis, making chemotherapy the main treatment option for these patients [6]. Chemotherapy is an anticancer therapy aimed at driving cancer cells to cell growth, arrest, or death [7]. One or several types of drugs are needed to impede the proliferation of cancer cells or other fast-growing cells, which can be achieved by many mechanisms, such as interfering with DNA synthesis. The first-line chemotherapies for patients with unresectable PC are 5-fluorouracil (5-FU) therapy and gemcitabine therapy [8]. However, due to chemoresistance, the prognosis of patients with PC has not been significantly improved with these treatments [9]. Gemcitabine is a deoxycytidine nucleoside analog that decreases cancer cell proliferation by inhibiting DNA synthesis [10]. Many mechanisms can explain gemcitabine chemoresistance, as widely reviewed in [11,12,13]. For example, cancer cells can reduce the expression of nucleoside transporters to prevent gemcitabine from passing through the cellular membrane [14]. Additionally, PC cells can also reduce the production of dFdCTP by inhibiting the expression of deoxycytidine kinase (dCK), the key enzyme involved in the transformation of gemcitabine into active gemcitabine triphosphate (dFdCTP) [10]. Furthermore, gemcitabine can induce the epithelial–mesenchymal transition (EMT), activate prosurvival signaling pathways, change cellular metabolism, and enhance cancer cell DNA repair mechanisms [15,16,17]. Altogether, these mechanisms may participate in the onset of gemcitabine resistance in PC. 5-FU is a pyrimidine analog that inhibits thymidylate synthase (TS), and its metabolites incorporate into RNA and DNA to induce cancer cell growth arrest and death [18]. Additionally, several mechanisms leading to the resistance of 5-FU-treated PC cells have been described, including the inhibition of the transport system of 5-FU into cancer cells and the mutation of key metabolic enzymes of 5-FU, such as 5-fluorodeoxyuridine monophosphate [18]. In conclusion, the mechanisms of chemoresistance in PC are still not fully explained, but according to recent studies, mitochondria may play important roles in the chemoresistance of PC.

## 2. Roles of Mitochondria in the Origin of Pancreatic Cancer

As several authors have extensively reviewed the basic physiology of mitochondria, we will summarize the role of mitochondria in pancreatic cells that can evolve into pancreatic tumors.

It is well documented that the secretory processes in both acinar and ductal cells require substantial energy consumption, which is produced by ATP hydrolysis (Figure 1). The energy demand of this process is mainly supported by oxidative phosphorylation [19,20]; indeed, intact functioning mitochondria are mandatory in the pancreas. In particular, acinar cells have highly specialized mitochondria with a novel intracellular location. There are three groups of mitochondria in pancreatic acinar cells: the perinuclear, perigranular, and subplasmalemmal subgroups [21,22]. The different subgroups play different roles in the fine-tuned regulation of cellular Ca^2+^ homoeostasis, which is directly associated with mitochondrial function because Ca^2+^ controls the activity of the rate-limiting enzymes of the tricarboxylic acid (TCA) cycle [23], which helps mitochondria adapt to the increased cellular ATP demand characteristic of the high secretory activity of acinar cells [23,24]. Furthermore, Ca^2+^ cross-talk with mitochondria produces ROS and controls the mitochondrial membrane potential. Other reports have indicated that acinar cells are supported by sustained glycolysis, as indicated by the observation that these cells exhibit oxidative and glycolytic metabolic functions in approximately equal amounts [20]. On the other hand, the fundamental roles of mitochondria in acinar cells are evident upon the induction of experimental acute pancreatitis in which impaired ATP production, defective autophagy, inflammatory responses, and necrosis are closely linked with a reduction in mitochondrial function caused by mitochondrial permeability transition pore (MPTP) opening upon mitochondrial Ca^2+^ overload (Figure 1). Indeed, under physiological conditions, the accumulation of Ca^2+^ in mitochondria does not change mitochondrial potential (ΔΨ); thus, preventing MPTP opening and highlighting the essential role of mitochondria under physiological conditions [25,26,27,28,29]. Other evidence of the important roles of mitochondria in pancreatic acinar cells is apparent from the observation that these cells rapidly accumulate glutamine in the cytoplasm [30,31], which is used in part for further substrate channeling into the TCA cycle and in part to synthesize glutamate that has been secreted in the pancreatic juice. In this regard, acinar cells express high levels of glutaminase (GLS) and glutamic dehydrogenase (GDH) enzymes, which are both required for glutamate synthesis and to produce TCA cycle substrates [32]. Importantly, acinar cells have great metabolic plasticity as indicated by their ability to accumulate glutamine to be metabolized into the TCA cycle for energy generation and to synthesize glutamate for secretion despite severe protein restriction in vitro and in vivo. Notably, under these experimental conditions, both GLS and GDH are transcriptionally enhanced to participate in acinar cell metabolic regulation [32].

In PDE cells, the function and subcellular distribution of mitochondria are not well characterized. Recently, an electron microscopic study of guinea pig PDE cells showed that the highest density of mitochondria is located in the apical part of the cells [33]. The function of this mitochondrial localization is not clear to date. However, it has been hypothesized that these mitochondria provide energy for ion secretion (bicarbonate) as mitochondria are localized mostly to the apical region of the cells [33]. Significantly, uncontrolled Ca^2+^ release in PDE cells can lead to intracellular Ca^2+^ overload and toxicity, leading to mitochondrial damage and impaired ATP production [34]. Indeed, recent findings suggest that the most common pathogenic factors leading to acute pancreatitis can evoke different types of intracellular Ca^2+^ signals, which can interfere with ductal cell function through alterations in mitochondrial activity [35]. The role of glutamine in PDE cells is also not clear. However, in vitro experiments using human pancreatic ductal epithelial cells (HPDEs) have shown a higher level of GLS expression and glutamine secretion in these cells than in cells derived from experimental pancreatitis or PC cells, suggesting that glutamine is used mainly as a substrate for glutamate synthesis and/or TCA cycle fueling and not for glutamate secretion, as observed with acinar cells [36]. Altogether, these observations underline an important role of mitochondria in normal acinar and duct cell physiology (Figure 1).

Another important metabolic characteristic of pancreatic cells is autophagy as an essential homeostatic process that maintains pancreatic acinar cell function. Indeed, in several models of pancreatic dysfunction, it has been observed that basal autophagy preserves the high rates of protein synthesis that characterize the exocrine pancreas by preventing endoplasmic reticulum (ER) stress, reactive oxygen species (ROS) accumulation, DNA damage, and mitochondrial dysfunction. In particular, it has been shown that proteins involved in mitophagy are necessary to eliminate dysfunctional mitochondria and thus prevent ATP depletion and regulate injury to acinar cells, preventing increased apoptosis and necrosis rates [26].

The relevant role of glucose metabolism, particularly mitochondrial activity, in β cells has been vastly discussed elsewhere [37]. However, as these cells do not originate from PC, we would only underline that inhibition of mitochondrial metabolism in β cells completely blocks insulin secretion and hence their main pancreatic function [38,39].

In summary, several findings indicate that mitochondria are essential organelles for pancreatic cell homeostasis and directly participate in the great pancreatic metabolic plasticity necessary to support its main functions.

### 2.1. Pancreatic Mitochondrial Dysfunction: From Pathological Pancreatic Conditions to Pancreatic Cancer

Diabetes mellitus type 2, acute and chronic pancreatitis (AP and CP), and inflammatory processes induced, for instance, by smoking are risk factors for PC. Pancreatitis is an interesting model in which to analyze the metabolic changes associated with PC. Indeed, as reviewed in [40], CP and PC share many features, such as fibrotic tissue, inflammation, genomic instability, and *K-RAS* mutations. Interestingly, *K-RAS* mutations, found in almost all cases of PC, have also been observed in approximately 30% of samples taken from patients with CP [41,42,43]. A direct association between *K-RAS* mutation in CP and PC onset has been established in animal models of human pancreatitis engineered to express a high level of an oncogenic form of *K-RAS*. These animals develop pancreatic intraepithelial neoplasia (PanIN) and invasive PC with clear evidence of inflammation and genomic instability [44,45]. On the other hand, CP tissues also generally contain early PanIN that often carries mutated *K-RAS* [46,47]. Similarly, inducible activation of the Ras pathway in mice can trigger alterations in acinar cells, provoking acinar-to-ductal metaplasia (ADM) and PanIN lesions, especially in a background of fibrosis and inflammation, further confirming the leading role of K-RAS in PC development and maintenance [48]. Interestingly, mitochondrial dysfunction, as previously described, is a characteristic of pancreatitis (Figure 1). Indeed, mitochondrial dysfunction occurs in both acinar and ductal cells [49], underlining the roles of mitochondria in the onset of pancreatitis. Furthermore, mitochondrial impairment in pancreatitis, especially that caused by Ca^2+^ overload, leads to alterations in glutamate-dependent metabolism; increased production of ROS; mitochondrial fragmentation; and perturbation of ATP-synthase activity, lipid metabolism, and autophagic pathways, which eventually lead to apoptosis first and then to necrosis, with the latter being more obvious in severe pancreatitis [26,50,51,52].

As changes in mitochondrial functions and *K-RAS* mutations in normal pancreatic cells are associated with pancreatitis and PC risk, a specific goal in recent years has been answering the question of whether mitochondria also play these roles in their malignant counterparts and whether K-RAS causes mitochondrial dysfunction [53,54]. In response, a recent work showed that *K-RAS* mutation alters mitochondrial metabolism in pancreatic acinar cells, resulting in increased production of mitochondrial ROS driving dedifferentiation of acinar cells to a duct-like progenitor phenotype and progression to PanIN. This process is mediated by complex signaling involving different players, such as the ROS-responsive kinase protein kinase D1 and the transcription factors NF-κB1 and NF-κB2, which upregulate the expression of the epidermal growth factor receptor and its ligands. Indeed, an in vivo reduction in K-RAS-mediated mitochondrial ROS generation reduces the development of preneoplastic lesions [55,56,57]. Furthermore, for a different cell model, the mouse embryo fibroblast (MEF) model, it has been reported that *K-RAS* mutated copy gain, proceeding from the *KRASG12D/WT* to a *KRASG12D/G12D* genotype, drives gains of function that include the upregulation and reprogramming of glucose metabolism, increased TCA cycle function, enhanced ROS regulation, and increased tumorigenic potential [58]. Additional connections between mitochondria, mutated K-RAS expression, and the appearance of early pancreatic lesions, such as ADM, have also been shown in K-RAS-mutant acinar cells. In particular, an early alteration in intracellular acetyl-CoA metabolism was shown with in vitro and in vivo models. Indeed, compared to normal acinar cells, differences in either substrate utilization or acetyl-coenzyme A (acetyl-CoA) levels were observed in both models. Specifically, K-RAS-mutant acinar cells show acetyl-CoA that is derived not only from leucine but also from glucose and palmitate, in contrast to normal acinar cells, suggesting different mitochondrial processes of acetyl-CoA generation. Importantly, this difference has also been observed in the context of PC derived from pancreatitis in which a *K-RAS* oncogenic mutation was present [59].

Another risk factor for PC is diabetes mellitus type 2. However, the association between sugar metabolism alteration and PC onset has not yet been fully elucidated [60]. Recently, it has been shown in the nontumorigenic human HPDE and HPNE pancreatic cell lines [61] and in mouse primary acinar cells that high-glucose culture enhances genomic instability, causing de novo oncogenic *K-RAS* mutations. In particular, a reduction in glucose utilization through glycolysis and the TCA cycle, favoring hexosamine pathway flux, has previously been described as a possible driver of cell transformation [62], and enhancing total protein glycosylation, in particular the ribonucleotide reductase enzyme, causes an imbalance in dNTP pools, accelerates mutagenesis and the selection of K-RAS-mutated cells and therefore accelerates cell transformation and pancreatic tumors [63].

Collectively, these data, obtained with different PC models, establish an association between oncogenic *K-RAS* expression, mitochondrial alterations, and multistep pancreatic tumorigenesis. However, although these findings suggest a picture in which mitochondrial changes play the role of driver for PC onset, pancreatic tumor cells, as described by numerous authors [48,64,65], prefer to take up more glucose and glutamine for energetic and anabolic purposes by enhancing glycolysis and glutaminolysis. Most cancer cells use these metabolites to provide cellular energy by a high rate of glycolysis after the cytosolic lactic acid fermentation, rather than by a somewhat low rate of glycolysis, after oxidation of pyruvate into mitochondria, as occurs in most non-cancerous cells, which is called the Warburg effect [66]. Indeed, glycolysis enhancement, even in the presence of abundant oxygen and glutamine utilization upon oncogenic K-RAS expression, surely plays a key role in the metabolic reprogramming observed in PC cells, ultimately impacting mitochondrial function.

Accordingly, in PC cells and mouse models, the oncoprotein K-RAS stimulates glucose uptake and channeling of glucose intermediates into the hexosamine biosynthesis pathway (HBP), pentose phosphate pathways (PPP), and serine biosynthesis pathway, thus promoting the glycolytic phenotype [48,67,68]. In this regard, gene expression and metabolic flux analyses show that oncogenic *K-RAS* upregulates the expression of glucose transporters to increase glucose influx and hexokinase (HK) 1 and 2, as well as pyruvate kinase M2 (PKM2), to accelerate glycolytic activity [48,67,69]. Importantly, this phenotype is enhanced by the extreme hypoxia found in PC, sometimes reaching approximately 20% of the pancreatic tumor [70]. In particular, HBP, driven by oncogenic *K-RAS* expression and hypoxia, provides precursors for protein *N*- and *O*-glycosylation, whose involvement in cancer proliferation, survival, and metastasis, and, almost as importantly, cancer cell metabolism has been recently and widely shown [62,70,71]. In addition, the enhanced entry of glucose carbon into the PPP, induced by oncogenic *K-RAS* and favored by negative HBP regulation of phosphofructokinase 1 (PFK1) through *O*-glycosylation [72], provides substrates for DNA and RNA synthesis under normal conditions of tumor growth and upon chemoresistance onset [73,74,75], reducing the importance of increased NADPH, which is necessary for the synthesis of fatty acids and glutathione (GSH), as well as for cell redox homeostasis [76]. Importantly, PPP activation has also been associated with metastatic PC [77]. Serine biosynthesis actively contributes to the generation of pyruvate and nonessential amino acids glycine and cysteine, which are used for purine nucleotide bases and GSH synthesis, respectively, as well as sphingolipid and phospholipid synthesis. Additionally, serine supplies carbon to the one-carbon pool, which is involved in the folate metabolism necessary for thymidine and purine synthesis and for the generation of S-adenosylmethionine (SAM), the methyl donor for both DNA and histone methylation reactions that influence DNA epigenetic gene expression processes [68,78,79,80] (Figure 3A).

In addition to changes in glucose metabolism in PC cells, changes in glutamine metabolism are also essential in tumorigenesis and growth. Indeed, glutamine, a multifunctional amino acid, is involved in protein synthesis and, as an anaplerotic substrate, in the TCA cycle, particularly when glycolytic pathways are enhanced. In addition, glutamine is also the donor of the nitrogen needed for the biosynthesis of purines, pyrimidines, NAD, asparagine, and UDP-N-acetylglucosamine via its terminal amide group. Glutamine also drives the uptake of essential amino acids, facilitates the recycling of excessive ammonia and glutamate, and regulates redox homeostasis [81,82]. Oncogenes can control and regulate glutamine metabolism in cancer cells [83]; in particular, a relevant role has been effectively described for oncogene *K-RAS* [36,65,84]. As glutamine is an important substrate for tumor, cancer cells often take up external glutamine more rapidly and in greater amounts than normal cells [85]. In addition, especially in highly glycolytic cancer cells, mitochondria are important sites of glutamine metabolism, as many important enzymes in this metabolic pathway, such as glutamic-oxaloacetic transaminase (GOT2), glutaminase (GLS1), and glutamate dehydrogenase (GLUD), localize to mitochondria [86]. As previously described, glutamine in cancer cells is a significant source of both reduced nitrogen for biosynthetic pathways and carbon for the TCA cycle [84]. However, in contrast to most cancer cells, where glutamine is converted to glutamate and then to α-ketoglutarate (αKG) through the catalysis of GLUD1, in PC, glutamine is mitochondrially converted to aspartate by GOT2 and then to oxaloacetate (OAA) as catalyzed by cytoplasmic aspartate transaminase (GOT1). Then, OAA is converted into malate by malate dehydrogenase (MDH1) and subsequently oxidized by malic enzyme (ME1) into pyruvate and the reducing power of NADPH [65]. This unique mitochondrial/cytoplasmic pathway is regulated by K-RAS, as it inhibits GLUD1 expression and stimulates GOT1 expression [65] (Figure 3A). Furthermore, the inhibition of different glutamine-catabolizing enzymes in PC cells has shed new light on glutamine metabolism in PC cells, identifying different key enzymes that, under normal or harsh growth conditions such as glutamine depletion or hypoxia, are necessary for PC growth. For instance, glutamate-ammonia ligase (GLUL), which catalyzes the synthesis of glutamine from glutamate and ammonia in an ATP-dependent reaction, is essential for cancer cell proliferation when the amount glutamine is limited, allowing cells to utilize other components, such as aKG and ammonium, to synthesize glutamine and support cell proliferation in PC [84]. Importantly, in contrast to GLS, whose inhibition does not have a significant effect on PC development in vivo [87], GLUL ablation causes a significant delay in disease progression, which is associated with decreased *O*-glycosylation and cell proliferation, underling the importance of the HBP in PC. On the other hand, it is also an important downstream glutamine metabolic pathway that is later activated under hypoxia in PC cells and benefits their survival. Indeed, under this growth condition, glutamine fructose-6-phosphate amidotransferase 1 and 2 (GFPT1 and 2), the HBP rate-limiting enzymes, are significantly upregulated, especially GFPT2 [70]. In addition, it has also been shown that GOT2 knockdown may lead to the induction of PC cell senescence [88]. The essential role of glutamine metabolism in *K-RAS* PCs was further assessed by *K-RAS* knockdown. Indeed, the inhibition of K-RAS significantly reduced and increased GOT1 and GLUD1 expression, respectively [65].

Altogether, glucose and glutamine metabolism rewiring in a K-RAS-dependent fashion almost completely changes normal mitochondrial function as K-RAS drives either gain or loss of metabolic routes, compared to normal pancreatic cells, with supportive roles in tumor development and progression.

### 2.2. Mitochondria in Cancer Cell Apoptosis

In addition to ATP synthesis and other anabolic processes, as delineated herein, mitochondria play an important role in controlling cellular apoptosis. Apoptosis is programmed cell death that removes unwanted cells through a complex signaling system in which mitochondria play essential roles [89]. The apoptotic process is activated by different pathways, including the death receptor-mediated or extrinsic pathway, the mitochondrial intrinsic pathway, and the ER pathway. All three pathways may converge first at mitochondria and then at the point of caspase cascade activation, which involves caspases 8, 9, 3, 6, and 7, as supporting players in the protein activation and degradation processes associated with the execution of apoptosis. In addition, another apoptotic pathway involves T cell-mediated cytotoxicity that activates the caspase cascade through perforin-granzyme-dependent activation. In addition to T cell-mediated cytotoxicity, proapoptotic and antiapoptotic members of the BCL-2 family, which are targets of external mitochondrial stimuli that induce apoptosis, mainly control this process (Figure 2A). In particular, the BCL-2 family comprises two proapoptotic groups, the BAX-like subgroup (BAX, BAK, and BOK) and the BH3-only subgroup (BAD, BIK, BID, BIM, BMF, HRK, NOXA, and PUMA), and an antiapoptotic group comprising BCL-2, BCL-xL, and Mcl-1 [90,91]. BCL-2 family proteins interact with each other or with other proteins through the BH-3 domain [92]. These proteins can determine whether the cell is committed to apoptosis or to survival by regulating cytochrome c (CYT-C) release from mitochondria through the control of outer mitochondrial membrane permeability. Indeed, a change in the outer mitochondrial membrane results in the opening of the mitochondrial permeability transition (MPT) pore, loss of the mitochondrial transmembrane potential, and release of the two mitochondrial proapoptotic protein groups, normally sequestered in the space between the inner and outer mitochondrial membranes, into the cytosol. The first group consists of CYT-C, a second mitochondria-derived activator of the caspase/direct inhibitor of apoptosis proteins (IAPs) binding protein with low pI (SMAC/DIABLO), high-temperature requirement/omi stress-regulated endoprotease, and serine protease omi protein A2 (OMI/HTRA2). In the cytosol, these apoptosis-related proteins initiate cell death by promoting caspase activation [92]. In particular, upon release CYT-C interacts with apoptotic protease activating factor 1 (APAF-1) and caspase 9 to form the apoptosome complex that triggers caspase-3 activation and cell death. In addition, the SMAC/DIABLO and OMI/HTRA2 proteins favor caspase activation by neutralizing the activity of endogenous IAPs [92] (Figure 2A). The second group of proapoptotic proteins, released in part from mitochondria during a late event that occurs after the cell has committed to apoptosis, comprises apoptosis-inducing factor (AIF), endonuclease G (ENDOG), and caspase-activated DNAse (CAD), which cause DNA fragmentation in a caspase 3-independent and caspase 3-dependent fashion, respectively (Figure 2A).

The intrinsic pathway is the main mechanism of apoptosis in mammalian cells; however, this pathway can also be activated by the extrinsic pathway through the protein BH3-interacting domain death agonist (BID), a proapoptotic member of the BCL-2 protein family that is translocated to mitochondria where it initiates the intrinsic pathway. In addition, the intrinsic pathway can also be activated by ER stress caused, for instance, by the accumulation of unfolded proteins, glucose shortage, or other types of stress that may culminate in the release of calcium from the ER into the mitochondria and thus initiate CYT C release (Figure 2A).

The suggestion that PC apoptotic machinery dysregulation is involved in the resistance to apoptotic stimuli and reduced sensitivity to therapeutic agents is currently debated. For instance, analyses of the proteins involved in apoptosome formation and function in several PC cell lines indicated that both the function and expression of these proteins, namely, APAF-1; caspase-3, -6, -7, -8, and -9; IAPs; SMAC; and Cit-C, are present in all cell lines [93]. However, comparative analyses of caspase-3 and caspase-8 activities and IAP protein expression levels established that, in PC cells, high basal caspase activity compared to normal pancreatic cells is counteracted by the upregulation of IAP expression, leading to survival [94,95]. Some tumors respond to apoptotic stimuli by increasing the expression of BCL-2. However, in pancreatic cancer cells, BCL-2 is expressed at normal or even decreased levels compared to normal pancreatic cells [95,96,97]. In contrast, BCL-xL overexpression has been detected in PC patients and has been associated with shorter patient survival times [98], accelerated PC carcinogenesis, and reduced senescence and apoptosis in PanIN, causing reduced survival time of K-RAS-mutant mice. Notably, BCL-xL deficiency induced senescence in PanIN cells [99]. Induced myeloid leukemia cell differentiation protein 1 (MCL-1) overexpression also reduces PC cell line sensitivity to chemotherapeutic drugs. Indeed, some PC cell lines exposed to 5-FU or gemcitabine show enhanced MCL-1 and/or BCL-xL expression and a higher ratio of these proteins to some proapoptotic proteins, such as BAX and BAK, suggesting that these proteins, especially Mcl-1, might be key players in the chemoresistance of PC [100]. In contrast, BAX and BAK upregulation in PC is associated with longer survival times and apoptosis in chronic inflammation regions surrounding the cancer cells, respectively [101,102] (Figure 2B).

In conclusion, deregulation of both pro- and antiapoptotic proteins plays an important role in PC growth and resistance to current therapy options.

### 2.3. mtDNA in Pancreatic Cancer Cells

The mitochondrial genome (mtDNA) is a maternally inherited 16.6 kb double-stranded, closed circular DNA that encodes two rRNAs, 22 tRNAs, and 13 proteins comprising various subunits of respiratory chain complexes. These subunits include seven subunits of complex I, one subunit of complex III, three subunits of complex IV, and two subunits of complex V. In addition, mtDNA contains a noncoding region located in the displacement (D)-loop involved in mtDNA replication and transcription [103].

Variations in mtDNA sequences have a greater impact on cell functions than nuclear DNA because mammalian cells contain thousands of mtDNA molecules; thus, mitochondrial genes are present in high copy numbers, and their products are essential for mitochondrial function. Moreover, both wild type and mutant mtDNA can coexist in cells in a condition known as heteroplasmy, and pathological functional mitochondrial impairment is associated with a change in the ratio between mutated and wild type mtDNA [104]. The threshold differs by tissues and according to the type of mutation, the value is estimated to be in the range of 60% to 90% mutant to wild type mtDNA [105]; notably, random mitotic segregation influences the proportion of mutant mtDNA inherited by daughter cells [106]. The accumulation of mutations in mtDNA is approximately tenfold greater than those in nuclear DNA. This higher level of mutations has been explained by a proposed “vicious cycle” mechanism in which mitochondria producing high levels of ROS induce mtDNA mutations. Then, these mutations affect the function of respiratory chain complexes that in turn cause further production of ROS, which induce other mtDNA mutations that are currently considered possibly responsible for aging and human diseases, including cancer [107]. Furthermore, the high frequency of mtDNA mutations seems to be a consequence of errors accumulated during replication (error-prone replication) in postmitotic tissues [108]. Indeed, the lack of histones and the inability of mtDNA repair systems to counteract the oxidative damage produced in mitochondria also contribute to the mtDNA mutation load [109,110]. Notably, mitochondrial genome instability is due to mutations that occur mainly in encoding genes but also in the D-loop. In particular, the latter is reported to be a mutational hot spot involved in carcinogenesis, especially in later cancer stages [111].

As mtDNA genes encode various subunits of respiratory chain complexes, it has been established that mtDNA mutations frequently lead to mitochondrial dysfunction and to changes in cellular metabolism, increasing the risk of cancer development. For instance, mtDNA mutations have been observed in cancer patients addicted to glycolysis-related pathways rather than oxidative phosphorylation (OXPHOS) [112]. As emphasized above, mtDNA changes are a consequence of germline and somatic mutations, as well as mtDNA copy number alterations. While different studies have demonstrated how these changes may be involved in the development of a wide range of human cancers [113], little information has been collected regarding PC association.

Nevertheless, some indirect and direct associations between mtDNA mutations and PC onset have been recently shown (Figure 3B). For instance, a mtDNA mutation analysis performed in 159 patients, of which 99 had confirmed PC, 42 had chronic pancreatitis, and 18 had pancreatobiliary tract tumors compared to 87 healthy subjects, indicated that approximately 16% of the patients affected by PC displayed mutations in the mtDNA D-loop region. In particular, a germline mutation (single-nucleotide polymorphism (SNP) at position 16,519), located in the D-loop region, was identified as a cooperating factor with the PC-associated diabetogenic factor involved in the onset of diabetes mellitus, a comorbidity frequently identified in PC patients [114]. This SNP has been associated with an alteration in the expression levels of some mitochondrial OXPHOS mRNAs in some pancreatic cells [115,116]. Another indirect association between mtDNA alterations and PC risk has been shown in a cohort of male smokers [117,118]. These subjects showed a significantly higher mtDNA copy number than nonsmokers, probably because of smoking-induced oxidative stress [119]. A more direct association between germline mtDNA mutations and PC risk has also been recently described. In a population-based case–control study of 532 PC cases and 1701 controls, some rare or low frequency variants, of which almost 70% occurred in genes codifying proteins involved in OXPHOS, such as NADH:ubiquinone oxidoreductase core subunit 4 and 5 (*ND4* and *ND5*) and cytochrome B (*CYT B*), or in the 12S RNA hypervariable and in some mitochondrial noncoding regions, were found. Importantly, both types of mutations, predicted to cause severe defects in OXPHOS, have been statistically associated with an increased risk of developing PC [120].

Regarding somatic mtDNA mutations occurring in tumors, a sequence analysis performed in different PC cell lines permitted the identification of approximately 226 homoplasmic variations, among which 23 occurred in coding regions. However, only one of these variants, affecting the mitochondrial *ND6* gene, had an association with the impairment of complex I activity in PC, while the effect of the other mutations has not been elucidated [121]. The need to identify other possible mtDNA mutations impinging on mitochondrial function in PC patients has recently led to the development of a more reproducible high-throughput sequencing tool (MitoChip) that has permitted the identification of mtDNA mutations from surgical resection specimens and fluid samples obtained from patients with pancreatic preneoplastic and neoplastic lesions [121,122]. By using MitoChip technology on samples derived from 15 PC patients, a study demonstrated that somatic mtDNA mutations are frequent in PC and that these mutations occur either in the noncoding region (44%) or in coding regions (56%), of which, for the coding regions, 25% are nonsynonymous (a nucleotide mutation that alters the amino acid sequence of a protein) and 31% are synonymous (a nucleotide mutation that does not alter amino acid sequences and is sometimes silent). Importantly, coding region mutations have been found in the *ND4*, *cytochrome c oxidase I* (*COI*), and *Cyt B* genes [123]. Unfortunately, the physiological consequence of these mutations has not yet been fully clarified. In contrast, a more recent report identified some mtDNA mutations and explained their effect on mitochondrial function. In particular, the authors, by using patient-derived PC cell lines, identified 24 somatic mutations in mtDNA, of which 87.5% were nonsynonymous and the remainder were synonymous. Strikingly, the vast majority of these mutations were localized in the genes encoding five subunits of complex I and two subunits of complex III (*CYT C oxidase I* and *CYT B)*. Moreover, by using a molecular modeling simulation, the authors suggested that these mutations can affect complex I stability, folding, and activity, and therefore can cause proton leakage as well as impairment of the redox capacity of CYT B subunits. Importantly, experimental data demonstrated that PC cell mitochondria with these mutations in complexes I and II were characterized by low oxygen consumption and the accumulation of succinate, confirming an association between mtDNA mutations and mitochondrial impairment [124].

Altogether, these findings suggest that mtDNA variations may contribute to tumor development and progression.

### 2.4. Mitochondrial Dynamics in Pancreatic Cancer

In recent years, increasing evidence has indicated a key role of mitochondrial dynamics, including fusion and fission, in driving malignant cancer phenotypes [125]. Mitochondrial fusion and fission are mechanisms through which mitochondria undergo constant changes in number and morphology, influencing the shape and functions of these organelles according to the metabolic status of the cells, and mitochondrial dynamics alter the metabolic state of the cells [126]. Among proteins involved in mitochondrial dynamics, dynamin-related protein 1 (DRP1), mitofusin 2 (MFN2), and optic atrophy protein family (OPA) GTPases are reported to have a key role in this process [127,128]. Importantly, their function is controlled by different mechanisms, such as transcriptional regulation, protein degradation [129], and post-translational modifications, among which a relevant role has been assigned to phosphorylation [130,131,132] (Figure 3C). For instance, it has been shown that changes in cytosolic Ca^2+^ levels, activating the Ca^2+^-dependent phosphatase calcineurin, lead to DRP1 dephosphorylation at Ser637, inducing DRP1 translocation to mitochondria and mitochondrial fragmentation [133]. In contrast, protein kinase A (PKA) activation by cyclic AMP, inducing DRP1 phosphorylation at Ser637, blocks its translocation and promotes mitochondrial elongation [134].

In addition, it has been shown that oncogenic K-RAS regulates the mitochondrial fission/fusion cycle. Indeed, both K-RAS mutant PC cell lines and resected tumor patients showed high levels of p-DRP1 on Ser616, mediated at least in part by K-RAS-dependent activation of ERK2 kinase. This phosphorylation causes mitochondrial fission in association with increased neoangiogenesis and tumor growth [135]. Importantly, other reports underlining the role of oncogenic K-RAS expression in favoring a fragmented mitochondrial phenotype [135,136,137] indicated that mitochondrial fragmentation may be avoided by the attenuation of oncogenic K-RAS signaling [136], DRP1 knockdown [135,137], and DRP1 PKA-dependent phosphorylation [138]. Notably, different ways to inhibit DRP1-dependent mitochondrial fission led to diminished in vitro and in vivo cancer cell growth, and in contrast PKA activation led to increased cancer cell survival under a profound glucose shortage. Notably, DRP1 phosphorylation by PKA induces mitochondrial interconnections, enhances complex I activity, increases mitochondrial glutamine utilization, reduces intracellular ROS generation, and stimulates autophagy [138,139], underlining the tight connection between mitochondrial dynamics and PC metabolism. In this regard, DRP1 loss in different K-RAS-driven PC models favors glycolysis and changes the levels of several TCA cycle metabolites (i.e., acetyl-CoA and αKG), induces the expression of hypoxia-inducible factor 1-alpha (HIF1α) and hexokinase 2 (HK2), impairs fatty acid oxidation, and induces mitochondrial respiration dysfunction [140]. In contrast, DRP1 overexpression in pancreatic cell and mouse models promotes tumor growth and lung metastasis, contributing to poor PC survival [140]. Another more indirect association between DRP1 and PC has been recently reported. In particular, it has been shown that in vitro and in vivo silencing of the family with sequence similarity 49 member B (FAM49B) protein, normally transcriptionally downregulated during the progression from early PanIN to adenocarcinoma, increases phosphor-DRP1 levels, causing mitochondrial fission, ROS production, and PPP activation. Notably, silenced FAM49B cooperated with oncogenic K-RAS in acquiring a cancer phenotype in a nontumor human pancreatic duct epithelial (HPDE) cell line, highlighting a synergic effect between K-RAS and FAM49B in tumor progression [141]. From a therapeutic point of view, it was recently shown that the pharmacological inhibition of DRP1 in PC cells and mice by treatment with Mdivi-1, a DRP1 inhibitor [142], induces mitochondrial fusion and decreases tumor growth and lung metastases. Moreover, Mdivi-1 treatment, which increases mitophagy, induces a significant reduction in mitochondrial mass, especially in the presence of oncogenic K-RAS, which eventually leads to a reduction in ATP production and oxygen consumption [143]. Altogether, these findings highlight the role of DRP1 in oncogenic Ras-driven transformation in PC, especially in cancer cells with abnormal fragmented mitochondria. In contrast, an oncosuppressive effect of the MFN2 protein in PC has also been widely described. Indeed, taking into consideration the ability of MFN2 to counteract DRP1 activity, as it is involved in mitochondrial fusion, it has recently been shown that MFN2 overexpression in AsPC-1 PC cells suppresses their proliferation, triggering apoptosis. Importantly, patients expressing high levels of MFN2 show an increased survival rate and a reduction in tumor aggressiveness [144]. In addition, other authors showed that MFN2 overexpression in KPC mouse-derived cells and in KPC mice reduces oxygen consumption, basal respiration, and ATP production compared with the effect in controls. Notably, such decreased OXPHOS activity correlates with reduced in vitro cell proliferation and in vivo tumor volume, as well as improvement in the survival of a syngeneic orthotopic mouse model of PC. Importantly, MFN2 overexpression, upon DRP1 inhibition, as previously described as causing prolonged mitochondrial fusion, induces the activation of mitophagy that eventually causes a significant reduction in mitochondrial mass and activity [143].

One of the main characteristics of PC is its highly heterogeneous architecture, with regions that may depend on particular metabolic pathways to adjust to the different environmental conditions. Indeed, PC tumors are often characterized by subpopulations addicted either to glycolysis or to OXPHOS [67,145,146]. Accordingly, it has been demonstrated that lipogenic PC cells are dependent on OXPHOS activity as a consequence of Myoferlin upregulation, which maintains mitochondrial organization by interacting with Mitofusin [147,148]. Indeed, Myoferlin silencing in PC cells, inducing phospho-DRP1-dependent mitochondrial fission, causes a reduction in mitochondrial activity that is partially compensated by the activation of autophagy, which is a mechanism to restore the energy needed for cell proliferation [149]. Confirmation of the role of Myoferlin in PC growth has been obtained by the finding that Myoferlin is upregulated in a cohort of 40 PC patients with low overall survival [149]. Surprisingly, in breast tumors, it has been shown that myoferlin is also involved in the processes of internalization and recycling of some receptors, such as the epithelial growth factor (EGF) receptor [150]. This observation is very important as EGFR is often activated in PC cells [151], contributing to tumor growth and gemcitabine resistance [71].

Taken together, these observations reveal a link between cellular metabolic changes and mitochondrial dynamics that can influence each other to sustain PC growth.

### 2.5. Pancreatic Cancer Stem Cells

PC stem cells (PCSCs) represent less than 1% of all PC cells and show high tumorigenic potential compared to noncancer stem cells (non-CSCs) [152]. Recently, CSC metabolism has attracted the attention of many researchers because it is involved in the ability of CSCs to maintain self-renewal, promote tumor initiation, and favor metastasis formation, enhancing both the proliferation and differentiation of surrounding cells. Moreover, metabolism plays a key role in the resistance to chemo/radiotherapy. For instance, transcriptomics and metabolomics analyses performed by using spheroids derived from five different PDX models showed the presence of three different spheroid subpopulations, each of which was characterized by a different metabolic profile (Figure 3D). In particular, the non-CSC group is glycolytic, the CSC group is oxidative, and the third group, also composed of CSCs, shows an intermediate metabolic profile. Notably, the latter type of CSC, expanded upon long-term treatment with the inhibitor of mitochondrial complex I metformin, shows high metabolic plasticity, i.e., the ability to switch between OXPHOS and glycolysis, which correlates with the high level of expression of MYC protein. Indeed, MYC protein, by downregulating peroxisome proliferator-activated receptor gamma coactivator 1-alpha (PGC1α) gene expression, induces diminishing mitochondrial activity that favors glycolysis and the accumulation of ROS, which at a high level negatively impacts their self-renewal ability [153]. Importantly, previous findings indicated that, compared to the non-CSC group and CSCs overexpressing MYC, the OXPHOS-addicted CSC group is highly sensitive to metformin because it expresses high levels of organic cation transporters 1, 2, and 3 (OCTI1-3), which are crucial for metformin cell uptake. Indeed, metformin treatment, causing ROS enhancement and decreasing the mitochondrial potential, leads to cell death [154]. Accordingly, metformin treatment in combination with gemcitabine reduces the in vitro sphere formation by human PC xenograft-derived cells [154], further suggesting an important role for mitochondrial activity in a subset of CSCs. The important role of mitochondria in several cancer cell models, including PC, has been confirmed upon treatment with dichloroacetate (DCA), an inhibitor of pyruvate dehydrogenase kinase (PDK). Indeed, DCA is able to reduce cancer cell survival [155,156,157], and, in particular, in the aggressive and chemoresistant PC cell line PANC-1, low doses of DCA altered their energetic metabolism and caused diminished stemness by decreasing either the ratio between CD24+/CD44+/EPCAM+ cells and CD24+/CD44+ cells or by affecting spheroid integrity and survival [158]. Another finding suggesting an important role of mitochondria in PCSCs may derive from the desmoplastic stroma limiting the glucose availability for cancer cells. In this regard, it has been observed that PC cells can stimulate autophagy in pancreatic stellate cells (PSCs). This process, causing the enhancement of protein catabolism in PSCs, generates alanine, which is secreted and then taken up by cancer cells. Importantly, alanine is converted into pyruvate, which fuels the TCA cycle, supports lipid and nonessential amino acid (NEAA) biosynthesis, and allows the redirection of glucose for serine and glycine biosynthesis. This cross-talk is also able to sustain tumor growth in harsh nutrient depletion conditions [159]. On the other hand, several other studies also associate PCSC addiction to both noncanonical glutamine metabolism and glycolysis. Indeed, either pharmacological or genetic inhibition of GOT1/2 [160] or GLUT1 [161] affects sphere formation ability and stemness and increases sensitivity to radiotherapy. In addition, secretome and proteomic analyses of PANC-1-derived CSCs showed increased expression of enzymes involved in glycolysis, PPP, the pyruvate–malate cycle, and lipid metabolism, suggesting that all these pathways may play key roles in CSC survival [162,163]. Indeed, MIA PaCa-2 and PANC-1 cell lines treated with the glycolytic inhibitor 3-bromopyruvate show a reduction in several stemness and pluripotency markers in association with a significant reduction in gemcitabine resistance, further suggesting that in both cell lines, the CSC subpopulations are dependent on glucose metabolism [164]. This important role of glycolysis in PCSCs has also been shown upon the onset of gemcitabine resistance. In particular, the metabolic characterization of a PC-resistant cell line (GR-Patu8988) generated after treatment with a high dose of gemcitabine indicated that these cells, showing features of CSCs and expressing different EMT markers [165,166], are also characterized by enhanced glycolytic flux and a reduction in intracellular ROS levels, which in turn negatively regulates the expression of doublecortin-like kinase 1 (DCLK1), a newly identified CSC marker critical for the maintenance of stemness and the EMT phenotypes in PC cells [167].

Collectively, this evidence demonstrates the key role of mitochondrial metabolism and of glucose and glutamine metabolism in the ability of CSCs to promote tumor development, invasion, metastasis and resistance to conventional chemotherapy (Figure 3D).

### 2.6. Cross-Talk between Pancreatic Cancer Cells and Stromatic Cells: Mitochondrial Metabolism in Perspective

PC cell metabolic rewiring is also supported by the stroma, particularly by pancreatic stellate cells (PSCs). PSCs, the predominant cell type in the pancreatic tumor stroma, are important mediators of the desmoplastic response, such as the induction of ECM remodeling, growth factor release, cancer cell motility and metastasis, and the emergence of the hypoxic environment characteristic of PC, which contribute to metabolic rewiring in both cancer cells and PSCs. In this regard, as previously described for pancreatic CSCs, it was recently shown that PC cells increase their oxygen consumption both in vitro and in vivo as a consequence of the major use of alanine-derived carbon released from PSCs (Figure 3E). Notably, in cancer cells alanine is used to fuel the TCA cycle to generate NEAAs and lipids. Furthermore, alanine availability permits the use of glucose for additional biosynthetic functions, such as the serine biosynthesis, and to reduce glutamine flux into the TCA cycle [159]. More recently, similar cross-talk connections affecting mitochondrial function were observed. Indeed, reciprocal signaling between PC cells and PSCs can restore the expression of some mitochondrial proteins in PC cells. This restoration induces an increase in mitochondrial respiratory capacity and re-establishes mitochondrial polarity and superoxide levels, processes normally reduced by oncogenic K-RAS in PC cells grown in the absence of PSCs [168]. This reciprocal signaling is also able to influence cancer cell metabolism through the activation of a PSC-dependent transcriptional program in cancer cells that enhances glycolysis, PPP, nucleic acid synthesis, and the TCA cycle. Notably, according to the changes in metabolic genes in PC cells, PSCs also start to consume more glucose and secrete more lactate, emphasizing the close metabolic relationship between the two cell types (Figure 3E). Additionally, it has also been shown that cancer-activated fibroblast (CAF)-derived exosomes containing intracellular metabolites, including amino acids, acetate, stearate, palmitate, and lactate, are able to inhibit mitochondrial oxidative phosphorylation, thereby increasing reductive glycolysis and glutamine carboxylation in cancer cells [169]. Collectively, these findings highlight the important role of PSCs in PC cell metabolism, sometimes with opposite effects on mitochondrial function, probably as a consequence of the dynamic alterations in the cancer environment, i.e., hypoxia and nutrient depletion, but suggest that the microenvironmental context broadly influences PC cell mitochondria.

## 3. Mitochondria-Driven Mechanisms of Drug Resistance in Pancreatic Cancer Cells

### 3.1. Antiapoptosis-Related Mechanisms in Mitochondria

The antiapoptotic mechanism is one of the main mechanisms of chemoresistance in cancer. The resistance of PC to gemcitabine treatment is also related to mitochondria-mediated apoptosis [170]. In particular, analysis of several kinds of PC cells expressing different levels of BCL-xL indicated that cancer cells with lower levels of BCL-xL have a higher sensitivity to gemcitabine [171]. Accordingly, PC cells induced to overexpress BCL-xL exhibit significantly enhanced resistance to gemcitabine [172]. Some proteins have been shown to influence chemoresistance in pancreatic cancer cells by regulating BCL-2 family proteins, such as RAB14. RAB14, a member of the RAS oncogene superfamily of small G proteins that regulates membrane vesicle transport and signal transduction, shows a close relationship with tumor proliferation and metastasis [173]. A transmembrane serine/threonine kinase (LMTK3) can phosphorylate a Rab effector (RCP) and improve the efficient invasion and metastasis in an in vivo model of PDAC. On the other hand, recent observations have indicated a role for RAB14 in chemoresistance. In PC cells, it has been shown that RAB14 overexpression reduces chemosensitivity to gemcitabine by increasing both mitochondrial membrane potential (MMP) and BCL-2 expression [171]. In contrast, the reduction in the RAB14 protein level, modulating either MMP or BCL-2 in opposite ways, enhances gemcitabine sensitivity. Notably, RAB14 positively regulates the expression of BCL-2 through the Hippo signaling pathway, which was previously shown to be associated with gemcitabine resistance [171].

The SMAC protein, as previously described, is a mitochondrial protein that can be released into the cytosol to inhibit the activity of IAPs [174]. Importantly, the activity of SMAC is also associated with PC cell sensitivity to chemotherapy [175]. Indeed, it has been shown that the SMAC mimetic SW IV-52 can overcome the chemoresistance of PC cells to gemcitabine by increasing cancer cell apoptosis through enhanced caspase activity [176]. In addition, XIAP expression can interfere with the effect of gemcitabine in PC cells, as knocking down XIAP levels in PC cells resensitizes cancer cells to gemcitabine [177].

In conclusion, the process of apoptosis of PC cells is closely associated with cancer survival and chemoresistance, and targeting apoptosis-related proteins can influence the chemosensitivity of PC cells to chemotherapy (Table 1).

### 3.2. Autophagy-Related Mechanisms in Mitochondria and Chemoresistance

Autophagy has been demonstrated to have tumorigenic functions because of its ability to support cancer cell survival [178]. In many cancer cell lines, K-RAS mutations can stimulate autophagy and enhance glycolytic capacity [178]. In recent years, scientists have found that proteins in mitochondria that are related to autophagy can increase the sensitivity of PC cells to gemcitabine [179]. Nutrient-deprivation autophagy factor-1 (NAF-1) is a protein located in the outer mitochondrial membrane where it plays an important role in regulating calcium, apoptosis, and autophagy [179]. NAF-1 belongs to a class of Fe-sulfur proteins containing a transmembrane domain necessary for mitochondrial integrity in mammalian cells [180]. In recent years, a study confirmed that NAF-1 was expressed in PC tissue and associated with the progression of PC [181]. The suppression of NAF-1 in cancer cells, which disrupts the Fe balance, can reduce cancer cell survival by affecting mitochondrial integrity and causing cancer cell death [180]. In this regard, it has been shown that the inhibition of NAF-1 in PC cells causes an inhibition of PC stem cell characteristics and their ability to invade, demonstrating the important role of NAF-1 in tumor progression [181]. Recent findings indicate that NAF-1 can interact with BCL-2 to regulate autophagy and apoptosis [182]. In particular, it has been shown that NAF-1 binds the BH3 and BH4 domains of BCL-2 and, in this way, may regulate autophagy and apoptosis. Indeed, BCL-2 can inhibit autophagy by interacting with the tumor suppressor BECLIN 1, and this interaction is also regulated by NAF1, which acts as an important cofactor of BCL-2 [183,184]. Recently, a study demonstrated the relationship between NAF-1 and the chemoresistance of PC cells [179]. In particular, inhibition of NAF-1 expression can enhance pancreatic cancer cell sensitivity to gemcitabine, demonstrating the function of NAF-1 in the chemoresistance of pancreatic cancer cells.

Mitophagy is another autophagic process that specifically eliminates damaged mitochondria. In cancer cells, mitophagy is associated with tumorigenesis and chemoresistance [185,186,187]. Notably, it has been observed that the oncogene KRAS can enhance the expression of BCL-2/adenovirus E1B 19 kDa protein-interacting protein 3-like (BNIP3L)/NIX and activate mitophagy in PC cells [188]. BNIP3L/NIX are mediators of mitophagy. Studies have demonstrated that the knockdown of BNIP3L/NIX impairs mitophagy, while the overexpression of BNIP3L/NIX can enhance mitophagy [189,190]. Enhanced mitophagy in PC cells restricts glucose flux to the mitochondria and drives glycolysis to promote PC progression [188]. On the other hand, it has also been shown that excessive mitophagy causing the loss of mitochondrial function in cancer cells may also induce cancer cell death and therefore reduce chemoresistance [191].

Altogether, autophagy and mitophagy can interfere with pancreatic cancer cell proliferation and mitochondrial function and hence regulate pancreatic cancer cell chemosensitivity (Table 1).

### 3.3. Cancer Metabolism and Chemoresistance

In recent years, scientists have gradually demonstrated that there is a direct correlation between anabolic glucose metabolism and drug resistance in PC cells [17,75]. Shangnan Dai et al. researched the relationship between glycolysis and gemcitabine resistance in PC cells [192]. They found that the expression of glycolysis-related genes such as HK1, glyceraldehyde 3-phosphate dehydrogenase (GAPDH), and PKM2 is closely associated with the prognosis of PC. Indeed, glycolysis inhibition in PC cells can significantly enhance the sensitivity of cancer cells to gemcitabine and improve the prognosis for patients [192]. Alice Nomura et al. focused on the chemoresistance of pancreatic CD133+ tumor-initiating cells (TICs) and found that CD133+ TICs have increased glucose uptake, which leads to increased glycolysis [193]. Shukla et al. found that the transmembrane glycoprotein Mucin 1 (MUC1), an oncoprotein playing a key role in tumor progression [194], upregulated the expression of HIF1α, which mediated increased glycolytic flux and pyrimidine biosynthesis in PC cells. This process can lead to gemcitabine resistance in PC cells [75]. As mentioned before, K-RAS plays an important role in glucose metabolism in PC cells by upregulating many key glycolytic enzymes, such as GLUT1 and HK, and the knockdown of these enzymes suppresses the proliferation and chemoresistance of PC cells [17]. Although more glucose is converted to lactic acid in cancer cells than in normal cells, mitochondrial respiration, as previously described, is also essential for PC proliferation [146]. For instance, a recent report focused on a subgroup of quiescent pancreatic tumor cells surviving KRAS oncogene ablation (hereafter, surviving cells (SCs)) identified, by using transcriptomic and metabolomic analyses, a significant enhancement of transcripts involved in mitochondrial function and an enhancement of mitochondrial function in these cells. Accordingly, SCs show higher oxygen consumption and reduced glycolysis than K-RAS-expressing cells. In addition, SC mitochondria were more hyperpolarized, elongated, and generated more ROS, further confirming major mitochondrial activity. Accordingly, these cells were significantly sensitive to the mitochondrial complex V inhibitor oligomycin as they were unable to activate a compensatory glycolytic mechanism compared to oncogenic K-RAS-expressing cells. Importantly, mitochondrial inhibition dramatically impacted the spherogenic and tumorigenic potential of SCs, leading to the prevention of pancreatic tumor relapse [146]. Mitochondrial complex I (NADH:ubiquinone oxidoreductase), which is involved in a key step of oxidative phosphorylation, the oxidation of NADH to NAD+, has also been demonstrated to contribute to chemoresistance [195]. In particular, complex I inhibition in PC cells can increase cancer cell sensitivity to gemcitabine [196]. In addition to energy metabolism, ROS metabolism may play an important role in the effect of chemotherapy resistance. Indeed, chemotherapy can alter ROS-related metabolic pathways and their intracellular levels, and it is widely believed that the anticancer effect of several chemotherapeutics is due to the oxidative stress in and ROS-mediated injury of cancer cells [197]. Creosote bush (*Larrea tridentata* (LT)) and green barley leaf extracts both attenuate H_2_O_2_-induced increases in ROS levels and inhibit the apoptosis of cancer cells, demonstrating the relationship of ROS metabolism and cancer cell survival [198,199]. However, the role of ROS in cancer cells is complex, as many studies demonstrate ROS as a tumor promotor, while many others show the opposite: ROS as a tumor suppressor [197]. Superoxide dismutase (SOD) is a mitochondrial protein that converts superoxide radicals to hydrogen peroxides, reducing the production of intracellular ROS. SOD plays an important role in antioxidant systems in mitochondria [176]. 2-Methoxyestradiol (2-ME) is a natural physiological estrogen metabolite that can induce cancer cell apoptosis by causing the accumulation of intracellular ROS. Scientists exposed PC cells to high concentrations of 2-ME to generate a cell line with resistance to 2-ME and high levels of ROS. They found that SOD was upregulated in this cell line, and suppressing the expression of SOD2 resensitized these PC cells to 2-ME, while the overexpression of SOD2 led to the opposite result, demonstrating a role for SOD in the PC cell chemoresistance to 2-ME [200]. Using PC cells, Qingcai Meng et al. showed that ROS can mediate the activation of AKT/glycogen synthase kinase 3β (GSK3β)/SNAIL signaling and contribute to gemcitabine resistance [201]. In addition, Girijesh et al. found that conditioned medium obtained from gemcitabine-treated PC cells can confer chemoresistance by contributing to a reduction in ROS levels by increasing the expression of catalase (CAT) and SOD2 protein (both ROS-detoxifying enzymes) [202]. Research on pancreatic tumor-initiating cells (TICs) shows that CD133+ pancreatic TICs protect them against the effect of chemotherapeutic agents by reducing the accumulation of ROS [193].

In addition to glucose metabolism, glutamine metabolism is an important target for improving PC cell chemosensitivity. As glutamine metabolism in PC cells is unique and important for cancer progression, many scientists focus on targeting glutamine metabolism to resolve the chemoresistance of PC cells. Hee Chan Yoo et al. found that a variant of the mitochondrial glutamine transporter SLC1A5, which is involved in glutamine transport into the inner mitochondrial membrane [203], is associated with gemcitabine resistance. Indeed, upon its overexpression or inhibition, chemoresistance in PC cells is enhanced and inhibited, respectively, demonstrating the role of glutamine metabolism in gemcitabine resistance. Suman Mukhopadhyay et al. found that KRAS can promote the expression of the transcription factor nuclear factor erythroid 2-related factor 2 (NRF2), which has been linked to the upregulation of glutamine metabolism in PC cells [204]. Disrupting the glutamine metabolic pathways in PC cells can significantly reduce their proliferation and improve their chemosensitivity [205]. Scientists have demonstrated that glutaminase inhibitors in PC cells can resensitize chemoresistant PC cells to gemcitabine [204].

The effects of mitochondrial metabolism and apoptosis on the sensitivity of PC cells to chemotherapy sometimes overlap with each other, as the alteration of the metabolic pathways in cancer cells also influences the ROS levels in mitochondria and then interferes with apoptosis cascades. For instance, as mentioned herein, the reduction in ROS levels in CD133+ TICs leads to the inhibition of apoptosis and increased resistance to chemotherapy drugs [193]. Furthermore, such metabolic rewiring also results in a higher expression of ATP-binding cassette (ABC) transporters, causing an increased efflux of chemotherapeutic agents and reducing the effect of chemotherapy [193]. However, further data need to be generated to better explain the association between metabolic rewiring and the ABC transporter function.

In addition, cancer metabolism can affect apoptosis through other pathways. The enzyme hexokinase (HK), which is located in the outer membrane of mitochondria, is a key enzyme in the glycolytic pathway. HK phosphorylates glucose and converts it to glucose-6-phosphate (G6P) for use in the metabolic pathway. Researchers have demonstrated that increased levels of HK2 in PDAC patients indicates a poor prognosis, especially because HK2 may induce tumor progression by stimulating lactate production in PC cells [206]. HK is upregulated in PC cells and is capable of inhibiting the apoptosis of cancer cells [17]. It has been reported that overexpression of HK2 (an isoform of HK) interacting with the voltage-dependent anion channel (VDAC) maintains the MMP and prevents the release of CYT-C, eventually inhibiting apoptosis [207]. Another enzyme, fructose bisphosphate aldolase (FBA), is also a metabolic enzyme that influences apoptosis when overexpressed in cancer cells. In particular, PDAC cells with highly expressed FBA, derived from fructose 1,6-biphosphate, show increased intracellular levels of glyceraldehyde-3-phosphate (G3P) and dihydroxyacetone, and a reduced cell death because of the inhibitory action of G3P on caspase-3 [208].

In conclusion, aerobic glycolysis, mitochondrial respiration, ROS metabolism, and glutamine metabolism all play important roles in tumor proliferation and contribute to the chemoresistance of PC cells (Table 1).

### 3.4. mtDNA and Chemoresistance in PC Cells

In addition to the apoptosis-related proteins and metabolism-related mechanisms that influence the resistance of PC cells to chemotherapy drugs, mutated mtDNA can also interfere with the sensitivity to chemotherapy drugs, and a lack of mtDNA causes increased resistance to therapeutic cancer agents [209]. mtDNA is shown to be closely associated with tumorigenesis and proliferation. Many clinical studies, as previously described, support the influence of mutated mtDNA in cancer cells [210,211,212]. In particular, recent findings support a direct link between a low mtDNA copy number, observed in some cancer cells, and a reduction in energy production through OXPHOS, which participates in the switch from mitochondrial respiration to the Warburg effect [213]. In fact, in some cases, mtDNA mutations leading to dysfunction of the electron transfer chain clearly cause OXPHOS inefficiency [213]. In addition to metabolism-related mutations, other findings indicate that the release of mtDNA can trigger inflammatory signals that may also induce the apoptosis of cancer cells [214,215]. Mitochondrial damage triggers the release of mtDNA, and then the cGAS/STING-mediated cytosolic DNA-sensing pathway can recognize it and induce interferon β (IFN-β) transcription and secretion, which can trigger NOXA expression in neighboring cells and activate apoptosis [216]. In recent years, an increasing number of studies have shown a relationship between mtDNA and the chemoresistance of PC cells [217]. Indeed, transfer of mtDNA cybrids from human PC cells to normal cells causes resistance to 5-FU and cisplatin in progeny cells [217]. Several mechanisms have been suggested to explain the relationship between mtDNA and chemoresistance in cancer cells [217,218]. For instance, mutations and reductions in mtDNA can enhance glucose uptake, which enhances tumor progression and chemoresistance to 5-FU in colorectal cancer cells [219]. In addition, mitochondrial microRNAs can induce chemoresistance in squamous cancer cells in the tongue by reprogramming tumor metabolism [220].

**Table 1 cells-10-00497-t001:** Related pathways and mechanisms of mitochondria-associated chemoresistance.

Physiological Process	Related Targets	Function on Chemoresistance	Reference
Apoptosis	BCL-xL	Inducing chemoresistance	[172]
	RAB14	Reducing chemosensitivity	[171]
	SMAC	Enhancing apoptosis and chemosensitivity	[175,220]
Autophagy and mitophagy	NAF-1	Regulating autophagy and inducing chemoresistance	[179,182]
	Mitophagy	Either promoting glycolysis and PC progression or causing the loss of mitochondrial function and chemosensitivity	[188,191]
Metabolism	Mitochondrial Complex I	Contributing to chemoresistance	[195,196]
	Glycolytic enzymes including HK and FBA	Inducing chemoresistance	[17,192,208]
	ROS	Mediating the activation of AKT/GSK3β/Snail signaling and contributing to gemcitabine resistance	[201]
	Glutaminase	Promoting tumor proliferation and chemoresistance	[204]
mtDNA-related	mtDNA mutations	Leading to the destruction of the electron transfer chain and OXPHOS inefficiency and enhancing glucose uptake and chemoresistance	[213,219]
	Low mtDNA copy number	Enhancing the switch from mitochondrial respiration to the Warburg effect	[213]
	Released mtDNA	Activating the cGAS/STING-mediated cytosolic DNA-sensing pathway that leads to IFN-β transcription and IFN-β secretion	[216]

## 4. Targeting Mitochondria to Overcome Chemoresistance

Mitochondria have been studied as anticancer targets for several years [221]. Many drugs that interact with different mitochondrial components show anticancer activity. As mitochondria are essential in the regulation of apoptosis, which is closely associated with the chemosensitivity of cancer cells, many drugs that target mitochondria aim to influence apoptosis to improve the outcomes of chemotherapy. As mentioned herein, BH3-only proteins are proapoptotic proteins in the BCL-2 family. In recent years, many studies have shown that BH3 mimetics can target BCL-2 family proteins to enhance apoptosis in cancer cells [222]. In recent years, scientists have found that BH3 mimetics may target at mitochondrial outer membrane permeabilization (MOMP) to enhance it and promote apoptosis in cancer cells [216]. Yi Zhou et al. demonstrated that in pancreatic cancer cells, BH3 mimetic ABT-199 can enhance the sensitivity of gemcitabine in vitro and in vivo [223]. Betulinic acid is a dietary agent derived from the outer bark of various plants and can induce apoptosis by triggering mitochondrial membrane permeabilization; betulinic acid has been shown to enhance PC cell death in combination with gemcitabine. In fact, this combination causes either loss of mitochondrial membrane potential or a decrease in PKM2, both of which favor PC apoptosis [224]. Furthermore, oridonin, a diterpenoid isolated from traditional Chinese medicine, can inhibit pancreatic cancer cell proliferation and synergize with gemcitabine to increase the apoptosis rate. This effect is due to BCL-2 expression inhibition and BAX expression induction [225]. Salinomycin is an antibiotic able to overcome ABC transporter-mediated multidrug resistance in human AML [226]. In addition, the apoptotic pathway activated by salinomycin is unique, as it is independent of p53 and caspase activation [227]. Guan-Nan Zhang et al. demonstrated that salinomycin can cooperate with gemcitabine to induce PC cell death. Specifically, salinomycin and gemcitabine enhance the apoptosis of CD133+ and CD133− cells, respectively, and consequently, their combination synergizes the induction of PC cell death [228]. Additionally, pretreatment with emodin, a tyrosine kinase II inhibitor, affects some mitochondrial apoptotic proteins, such as by inhibiting NF-κb and promoting BAX, and improves the sensitivity of PC cells to gemcitabine [229].

Many targets associated with metabolism in cancer cells have been studied to improve cellular sensitivity to chemotherapy [17,230,231]. As mentioned herein, HIF-1α mediates the increase in glycolytic flux in PC cells. K. Shukla et al. found that digoxin functions as an inhibitor of HIF-1α, and the combination of gemcitabine and digoxin can significantly resensitize PC cells to gemcitabine [75]. Recently, gemcitabine-dependent upregulation of the enzyme phosphoacetylglucosamine mutase 3 (PGM3) has been observed. In particular, treatment of PC cells and pancreatic PDX mice with gemcitabine induced PGM3 expression at the mRNA and protein levels. Higher PGM3 expression is associated with enhanced HBP flux, leading to an increase in N- and O-protein glycosylation. Strikingly, PGM3 inhibition, elicited by the specific inhibitor FR054, combined with gemcitabine strongly reduced the growth and survival of PC cells as well as tumor growth in both pancreatic xenograft and PDX mouse models [71]. Scientists used the drug phenformin targeting mitochondrial complex I to inhibit oxidative phosphorylation in PC cells and found that the sensitivity to gemcitabine was enhanced [196]. Similar results were obtained by using another biguanide, metformin [231]. Metformin, primarily used as a first-line drug in type II diabetes, may either modulate blood glucose levels by activating 5-AMP-activated protein kinase (AMPK) [232] or inhibit respiratory complex I, causing a significant decrease in ATP synthesis via oxidative phosphorylation [233]. In recent years, scientists have found that metformin can function as a tumor suppressor due to several mechanisms, including the inhibition of mitochondrial complex I, suppression of mammalian target of rapamycin complex 1 (mTORC1), and activation of autophagy [230,234]. Studies have shown that metformin can also be combined with chemotherapeutics in PC cells to enhance the drug effect [234,235]. However, the effect of metformin is not satisfactory when synergistically combined with gemcitabine in PC patients, as several randomized controlled trials showed that PC patients did not benefit from metformin [236,237]. In 2019, Broekgaarden et al. used patient-derived pancreatic CAFs and found that the complex I inhibitor metformin combined with oxaliplatin can overcome drug resistance of CAFs [238].

As mentioned herein, ROS levels in cancer cells are associated with chemoresistance [239]. Lipid membrane-coated silica-carbon (LSC) hybrid nanoparticles have been reported to target ROS production, leading to an increase in ROS levels in mitochondria through pyruvate. ROS can reduce the amount of ATP in cancer cells, and then the ATP-driven transmembrane efflux pumps that lead to chemotherapy drug efflux and drug resistance are inhibited, leading to the remission of cancer chemoresistance [240]. Uncoupling protein 2 (UCP2), a member of the mitochondrial uncoupling protein family, can reduce the proton gradient in normal cells [241]. In recent years, different reports have indicated that UCP2 is highly expressed in cancer cells, such as pancreatic and gallbladder cancer cells [241], where it participates in metabolism and ROS level regulation. Notably, in gallbladder cancer cells, the knocking down of UCP2 can improve the sensitivity of cancer cells to gemcitabine [242]. In particular, gemcitabine treatment induces a significant increase in ROS in UCP2-silenced cells compared to control cells, suggesting that ROS are involved in UCP2-mediated chemoresistance [242].

In addition to glucose metabolism, drugs targeting glutamine metabolism in PDAC are also a research hotspot. The glutamine analog 6-diazo-5-oxo-L-norleucine(DON) is effective in improving the chemosensitivity of PC cells [205]. Many drugs targeting glutamine metabolism enzymes have also been studied, especially glutaminase inhibitors. Scientists used buthionine sulfoximine (BSO), a glutamylcysteine synthetase inhibitor, to inhibit GSH synthesis (a vital part of glutamine metabolism) and found that gemcitabine sensitivity was rescued in PC cells [203]. In addition, the glutaminolysis inhibitors bis-2-(5-phenylacetamido-1,3,4-thiadiazol-2-yl)ethyl sulfide (BPTES) and CB-839 can cause the same effect in PC cells [203,204].

In recent years, many articles have demonstrated that mutated mtDNA may be a target for cancer therapy [243]. For example, use of mitochondria-targeted nucleases has been experimentally applied to some kinds of cancer cells [244]. Mitochondria-targeted nucleases are primarily used to cleave mtDNA and then to modify the oxidative phosphorylation system encoded by mtDNA. Particularly, this approach has been used to treat mitochondrial diseases, including neurodegenerative diseases such as Alzheimer’s disease [245]. However, recently, it has also been applied to cancer therapy [244]. Among these nucleases, mitochondria-targeted transcription activation-like effector nucleases (mito-TALENs) and zinc finger nucleases (ZFNs) are frequently used. Sandra et al. used Δ5–mito TALEN in human osteosarcoma cells and found that Δ5–mito TALEN significantly reduced the mtDNA deletion load and increased complex I activity in osteosarcoma cells [244]. Research on the use of mitochondria-targeted nucleases as treatments to resolve chemoresistance in PC is limited, but it undoubtedly is a hopeful research direction for the future. In addition to nucleases, mitochondrial miRNAs, which are also called mitochondrial microRNAs (mitomiRs), can influence the chemoresistance of cancer cells by regulating mitochondrial transcription. In tongue squamous cell carcinoma cells, a mitomiR regulates mitochondrial gene transcriptional activities and cisplatin sensitivity [220].

## 5. Conclusions

PC has a high mortality rate because of its difficulty to diagnose and its resistance to chemotherapy. Mitochondria show essential functions in cell biology, such as producing ATP and regulating apoptosis and autophagy, making mitochondria a hopeful target in cancer therapy. Moreover, researchers have found that mitochondria play important roles in the chemoresistance of PC by affecting apoptosis, metabolism, mtDNA, and other factors. Interfering with these factors in mitochondria can improve the sensitivity of PC cells to chemotherapeutic agents, such as gemcitabine, making mitochondria a promising target for overcoming chemoresistance in PC (Figure 4). In conclusion, mitochondria contribute greatly to chemoresistance in PC cells, and targeting mitochondria is a promising way to overcome drug resistance in PC.

## Figures and Tables

**Figure 1 cells-10-00497-f001:**
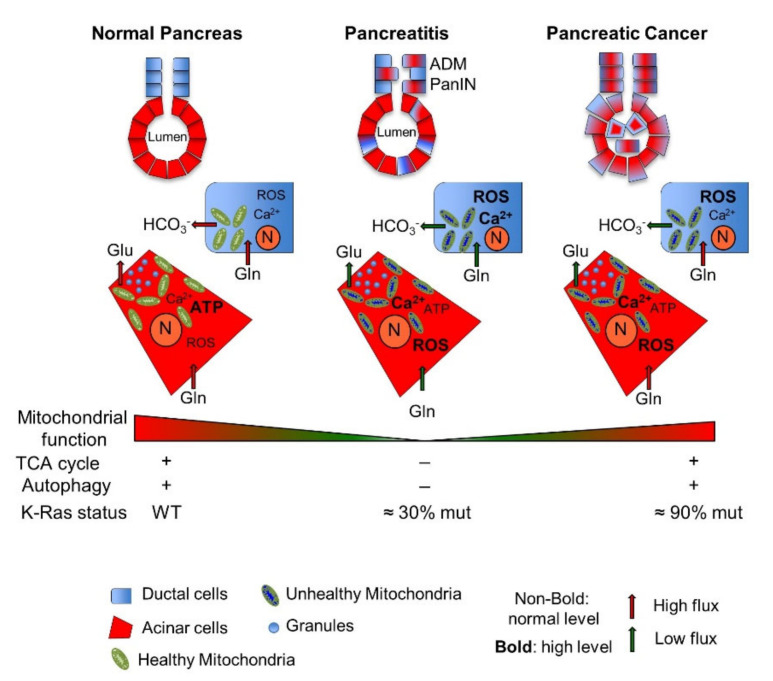
Schematic view of mitochondrial function and activity under physiological and pathological conditions in normal and cancer pancreatic cells.

**Figure 2 cells-10-00497-f002:**
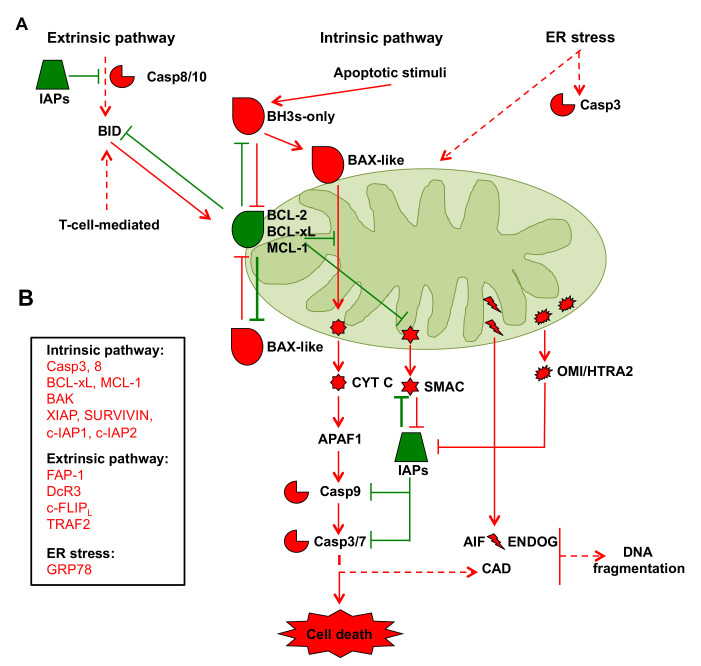
(**A**) Schematic view of apoptotic mechanisms. Green lines and shapes: antiapoptotic proteins and mechanisms; red lines and shapes: proapoptotic proteins and mechanisms. (**B**) Apoptosis-related proteins overexpressed in PC. Fas-associated phosphatase-1 (FAP-1), decoy receptor 3 (DcR3), Cellular FADD-like IL-1β-converting enzyme (FLICE)-inhibitory protein (c-FLIPL), TNF receptor associated factor 2 (TRAF2), glucose-regulated protein 78 kDa (GRP78).

**Figure 3 cells-10-00497-f003:**
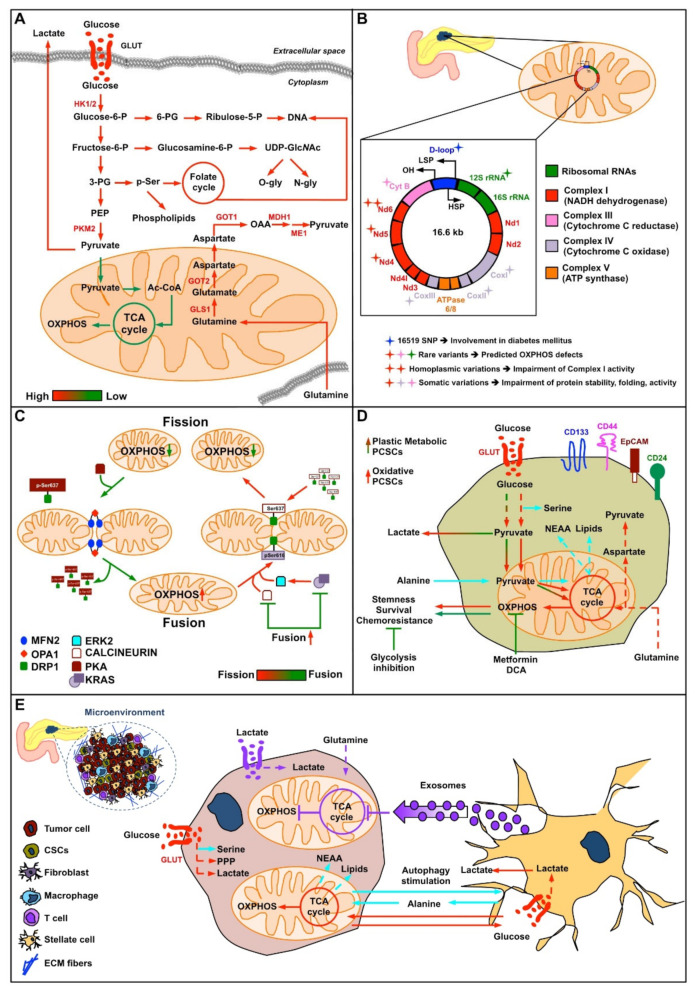
(**A**) Schematic representation of the main metabolic pathways in PC. Red indicates increased protein expression and/or metabolic flux; green indicates decreased metabolic flux. (**B**) Schematic representation of mtDNA and main mutations identified in PC. LSP and HSP arrows designate the promoter and direction of the H-strand (HSP) and L-strand (LSP), respectively. Colors refer to specific mtDNA-encoded OXPHOS subunits. Red: complex I subunits. Pink: complex III subunits. Purple: complex IV subunits. Orange: complex V subunits. Blue: D-loop region. Green: ribosomal RNAs. The colored stars indicate the mutations found or predicted in PC mtDNA. (**C**) Schematic representation of the mitochondrial fusion and fission cycle and some of the proteins involved in the regulation of this cycle. (**D**) Schematic representation of PCSC metabolism and some stemness markers. Red refers to an increased effect, green refers to a reduced effect, and green/red represents PCSC metabolic plasticity (for detailed information, refer to the main text). (**E**) Schematic representation of tumor cell composition and the identification of the main metabolic cross-talk connections identified between cancer cells (left) and stellate cells (right). The different colors (red, cyan, and purple) indicate the three cross-talk connections described in the main text.

**Figure 4 cells-10-00497-f004:**
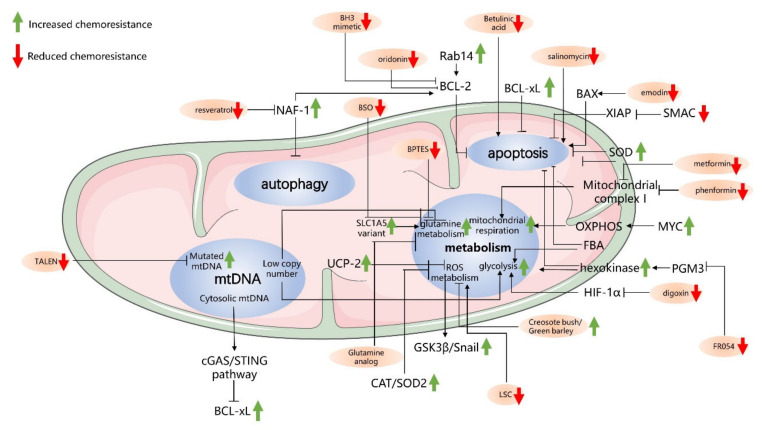
The mechanism of mitochondria-mediated chemoresistance in pancreatic cancer cells and ways to overcome it. In pancreatic cancer cells, mutated mitochondria induce chemoresistance via many mechanisms, including the inhibition of apoptosis, regulation of autophagy, altered metabolism, and mtDNA mutation. Many drugs and treatments that target these mutations are being studied to overcome chemoresistance in PC cells.

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
