# Peer review of "The Role of Mitochondria in the Chemoresistance of Pancreatic Cancer Cells"

_cells, 2021, doi:10.3390/cells10030497_

Round 1
Reviewer 1 Report
The article covers many important information regarding to the role of mitochondria in chemoresistance of pancreatic cancer. However, the contents are too trivial, some not-that-relevant information are also included, that will cause the principal axis of this paper be disperse. Trimming the article, decrease the volume of too basic part (eg. the introduction of PC; Part 2.2; Part 2.5; Part 3.1) is suggested. Reorganization of Table 1 is also recommended.
Reviewer 2 Report
The Authors responded all the comments accordingly!
The manuscript is ready for its publication.
Author Response
Thank you for your approval!
This manuscript is a resubmission of an earlier submission. The following is a list of the peer review reports and author responses from that submission.
Round 1
Reviewer 1 Report
The review "Mitochondria and chemoresistance in pancreatic cancer cells" is a really interesting topic since pancreatic cancer is becoming in several countries the most lethal type of cancer.The review is well structured and covers a good part of the literature, especially the older one. However, several parts should be expanded adding more information, for example paragraph 2, or divided, for example paragraph 2.3 must be divided into a paragraph describing the relationship between pancreatic cancer and mtDNA and another section that talks about pancreatic cancer and mitochondrial dynamics.
The Authors could add and analyze several other references such as:
Zhang G.N., Liang Y., Zhou L.J., Chen S.P., Chen G., Zhang T.P., Kang T., Zhao Y.P. Combination of salinomycin and gemcitabine eliminates pancreatic cancer cells. Cancer Lett. 2011;313:137–144. doi: 10.1016/j.canlet.2011.05.030.
Esumi H., Lu J., Kurashima Y., Hanaoka T. Antitumor activity of pyrvinium pamoate, 6-(dimethylamino)-2-[2-(2,5-dimethyl-1-phenyl-1H-pyrrol-3-yl)ethenyl]-1-me thyl-quinolinium pamoate salt, showing preferential cytotoxicity during glucose starvation. Cancer Sci. 2004;
Oncogene ablation-resistant pancreatic cancer cells depend on mitochondrial function. Nature. 2014; 514: 628-632
Yang S, Wang X, Contino G, Liesa M, Sahin E, Ying H, et al. Pancreatic cancers require autophagy for tumor growth. Genes Dev. (2011) 25:717–29. doi: 10.1101/gad.2016111
Liou GY, Döppler H, DelGiorno KE, Zhang L, Leitges M, Crawford HC, et al. Mutant KRas-induced mitochondrial oxidative stress in acinar cells upregulates EGFR signaling to drive formation of pancreatic precancerous lesions. Cell Rep. (2016) 14:2325–36. doi: 10.1016/j.celrep.2016.02.029
Yang S, Hwang S, Kim M, Seo SB, Lee JH, Jeong SM. Mitochondrial glutamine metabolism via GOT2 supports pancreatic cancer growth through senescence inhibition. Cell Death Dis. (2018) 9:55. doi: 10.1038/s41419-017-0089-1.
Furthermore, some acronyms are used without adding the full name (line 160 IAP to be moved to line 157), P-gp to line 249 what is it?
The manuscript can be accepted with major revision and several text editing.
Reviewer 2 Report
- Writing of the article is careless, there are inconsistent abbreviation words and the redundant sentences even in the Abstract. For example, the authors didn't mention what is "PC" in the very first sentence of the abstract, but use the full name "pancreatic cancer" in the end. Line 20 "mitochondria may play an important role in the chemoresistance of PC." and Line 21 " mitochondria play an important role in the chemoresistance of pancreatic cancer " shows high redundancy.
- Figures are not informative enough for elaborating the key concepts of this review article.
Reviewer 3 Report
The manuscript attempts to provide an overview of the role of mitochondria in cancer (focused on prancreatic cancer) chemoresistance. After a very short overview about therapies in pancreatic cancer, the authors introduced mitochondria cellular functions in metabolism and apoptosis, including a section dedicated to mtDBA and mitochondrial dynamics, before reporting mechanisms involved in drug resistance reying on apoptosis, autophagy or metabolism. Eventhough, the topic is of particular interest, this review is overall superficial and needs to be completed with recent findings included for example in the following publications that highlight the major role of mitochondria in cell death induced by chemotherapy, including the related molecular mechanisms (Bock and Tait, Nat Rev Cell Mol Biol 2020, Giacomello, Nat Rev Cell Mol Biol 2020, Mills E, Nature Immunology 2017). Cellular and molecular aspects of chemoresistance included in the review, often need more description to be well understood (for example role of BCL2 family proteins in apoptosis section (line 100) or SMAC function (line 160) that modulates caspase activity downstream mitochondria outer membrane permeabilization (MOMP) rather than upstream cyto-c release. The focus on the role/function of Rab14 (line 145) in chemoresistance should be described in more details to be convincing. Many publications now report that in addition to cancer-related mutations (with functional impact on cell metabolism), mtDNA when released from mitochondria triggers inflammatory signaling in cancer cells that modulate their response to anticancer treatment (Giampazolias , Nat Cell Biol 2017). In the last section, the use of BH3 mimetics already included in clinical practice in oncology, to target mitochondria (in triggering MOMP) in cancer cells should be mentioned (Lohard S, Nat Comms 2020). Many very interesting points points included deserve however to be described more precisely (mitochondria dynamics (lane 130), NAF1 (lane 169), MCL1 (as a BCL2 family protein (lane 183), TIC (lane 208)...
Overall, the manuscript needs to be deeply completed and corrected. More recent references have to be included and citated data or reports should be more accurately described to appear relevant to the topic. Expressions such as many other factors ((lane 31, 160) or and so on (lane 50) should be avoided. The figures could better support the text and help to illustrate the interactions between relevant cellular processes.
Reviewer 4 Report
Reviewer general comments:
The review “Mitochondria and chemoresistance in pancreatic cancer cells” by Fu et al. is an interesting and original topic; Manuscript ID: cells-901743. Moreover, the manuscript is well written and structured and deserves to be published in the Cells journal. However, this reviewer has some minor comments for the authors.
Authors:
Lines 46-47: “MtDNA is 16,569 47 bp in length…”
Reviewer comment 1:
The Authors should consider using the following portion: “The human mtDNA comprises 16,569 base pairs (bp) in length…”
Authors:
Lines 60-61: “The Warburg effect is associated with the altered mitochondria in cancer cells; for example, mtDNA-depleted cells exhibit the Warburg effect[15].”
Reviewer comment 2:
The Authors should include a more ample definition of the Warburg effect: “The Warburg effect is a state that most cancer cells use to provide cellular energy by a high rate of glycolysis after the cytosolic lactic acid fermentation, rather than by a somewhat low rate of glycolysis, after oxidation of pyruvate into mitochondria, as occurs in most non-cancerous cells [Liberti et al. 2016].”
Liberti MV, Locasale JW. The Warburg Effect: How Does it Benefit Cancer Cells? Trends Biochem Sci. 2016 Mar;41(3):211-218. doi: 10.1016/j.tibs.2015.12.001.
Reviewer comment 3:
If possible, in the section “4 Targeting mitochondria to overcome chemoresistance “, line 238, the authors could include the following:
“Exposure of cancer cells with plant extracts isolated from both green barley young leaves (Hordeum vulgare L.) or creosote bush (Larrea tridentata) exhibited significant protection under a potent H2O2-induced oxidative stress, as evidenced by improvement in the viability of the cancer cells (Ruiz-Medina et al. 2019; Moran-Santibañez et al. 2019).
Ruiz-Medina BE, Lerma D, Hwang M, Ross JA, Skouta R, Aguilera RJ et al. Green barley mitigates cytotoxicity in human lymphocytes undergoing aggressive oxidative stress, via activation of both the Lyn/PI3K/Akt and MAPK/ERK pathways. Scientific reports. 2019;9(1):1-11.
Morán-Santibañez, K.; Vasquez, A.H.; Varela-Ramirez, A.; Henderson, V.; Sweeney, J.; Odero-Marah, V.; Fenelon, K.; Skouta, R. Larrea tridentata Extract Mitigates Oxidative Stress-Induced Cytotoxicity in Human Neuroblastoma SH-SY5Y Cells. Antioxidants 2019, 8, 427.
Reviewer 5 Report
- The title of the article is suggested to be changed to a more meaningful one; for example “The role of mitochondria in the chemoresistance of pancreatic cancer”.
- I see the article needs significant language improvements. There are a number of typo errors throughout the manuscript.
- As this is a review article the introduction section must be improved. Authors can add more information about pancreatic cancer and recent advancements in pancreatic cancer with updated references. I encourage you to add recent references from reputed articles.
- The section 2.1, mitochondria in cancer metabolism, has also to be improved. Authors can find a number of articles related to mitochondria in cancer metabolism and update with a more detailed figure. The figure belongs to the section 2.1 does not fit to the section. It has to be improved.
Eg. Mitochondrial Metabolism as a Target for Cancer Therapy published in Cell Metabolism, 2020 (10.1016/j.cmet.2020.06.019)
- The section 2.2 related to the mitochondria in apoptosis is not informative. Authors can find a range of article relating to this and update with more references. Also, improve the figure.
- The title of section 3, Mitochondria and drug resistance in pancreatic cancer cells, has to be changed to a meaningful one. Also, all the information in the sections 3.1-3.4 need to be summarized in a table with references. You can add the table next to the section 3.
- The section 4 can be improved with recent references. Also, the identified targets is suggested to be presented in a figure or table.